# Crystallinity and Gas Permeability of Poly (Lactic Acid)/Starch Nanocrystal Nanocomposite

**DOI:** 10.3390/polym14142802

**Published:** 2022-07-09

**Authors:** Somayeh Sharafi Zamir, Babak Fathi, Abdellah Ajji, Mathieu Robert, Said Elkoun

**Affiliations:** 1Department of Chemical and Biotechnological Engineering, University of Sherbrooke, Sherbrooke, QC J1K 2R1, Canada; babak.fathi@outlook.com (B.F.); mathieu.robert2@usherbrooke.ca (M.R.); said.elkoun@usherbrooke.ca (S.E.); 2Department of Chemistry, University of McGill, Montreal, QC J1K 2R1, Canada; 33SPack, CREPEC, Chemical Engineering Department, Polytechnique Montreal, Montreal, QC H3C 3A7, Canada; abdellah.ajji@polymtl.ca

**Keywords:** poly (lactic acid), grafted starch nanocrystals (g-SNCs), crystallization, crystal structures, microstructure, permeability relations

## Abstract

The present work seeks to determine the impact of weight percentage (wt%) of grafted starch nanocrystals (g-SNCs) on the oxygen and water vapour permeability of poly (lactic acid), PLA. Changes in the oxygen and water vapour permeability of PLA due to changes in PLA’s crystalline structures and lamellar thickness were quantified. To this end, 3, 5, and 7 wt% of g-SNC nanoparticles were blended with PLA using the solvent casting method in order to study impact of g-SNC nanoparticles on crystallization behaviour, long spacing period, melting behavior, and oxygen and water barrier properties of PLA nanocomposites. This was achieved by wide-angle X-ray diffraction (WAXD), small-angle X-ray diffraction (SAXD), differential scanning calorimetry (DSC), and oxygen and water vapour permeability machine. The results of the WAXD and SAXD analysis show that the addition of 5 wt% g-SNC in PLA induces α crystal structure at a lower crystallization time, while it significantly increases the α crystal thickness of PLA, in comparison to neat PLA. However, when g-SNC concentrations were altered (i.e., 3 or 7 wt%), the crystallization time was found to increase due to the thermodynamic barrier of crystallization. Finally, the oxygen and water vapour permeability of PLA/SNC-g-LA (5 wt%) nanocomposite film were found to be reduced by ∼70% and ~50%, respectively, when compared to the neat PLA film. This can lead to the development of PLA nanocomposites with high potential for applications in food packaging.

## 1. Introduction

Polymeric materials are widely used in the food packaging industry because they are lightweight, have a relatively low cost of production, and possess good mechanical properties essential for the food packaging industry [1]. However, most of the polymeric materials currently being used are petroleum-based polymers, which are inherently non-biodegradable and non-renewable. Therefore, there is a major thrust towards development of a biodegradable and renewable alternative [2,3]. Among the various alternative biodegradable polymers known, Poly (lactic acid), PLA, is one the most promising due to its wide availability, ease of processing, suitable mechanical properties, and high transparency [4,5,6]. However, currently the use of PLA in the food packaging industry is limited due to its high gas permeability, specifically those of oxygen and water vapour [7,8]. This is because pure PLA has a higher amorphous phase compared to the crystalline phase, which is known to facilitate gas permeability [7,8]. Therefore, current research efforts are focused on improving the gas barrier properties by increasing the crystalline phase and thus reducing the amorphous phase present within PLA [7,8].

In addition to the ratio of crystalline to amorphous phase, the crystalline structures also play an important role in the gas permeability of PLA [9,10,11,12,13,14]. The crystalline structures and their morphology within a crystalline phase of PLA can be modified by subjecting it to thermal treatment [10,11]. Specifically, two different crystal structures of PLA exist, α and $\alpha^{\prime }$ structure. The difference in crystal structures depends on the thermal crystallization temperature [15,16,17,18,19]. The existence of PLA as a polymorphic material depending upon thermal treatment is a manifestation of the link between its thermal and physical properties. The α structure of PLA is known to exist above 120 °C crystallization melting temperature [17,18,19]. In contrast, the $\alpha^{\prime }$ structure is observed below 100 °C crystallization melting temperature [17,18,19]. A mixture of both crystalline structures (α and $\alpha^{\prime }$) was found between 100–120 °C [17,18,19]. The phase transition from $\alpha^{\prime }$ to α structure involves rearrangement of the molecular chains, which leads to a more packing order within a unit cell [19].

It has been shown that the gas barrier properties of semicrystalline polymers such as PLA are substantially modified by crystalline polymorphisms [20,21]. For example, Cooca et al. showed that the presence of a highly ordered α crystal structure enhances the barrier properties of Poly-l-Lactic Acid, PLLA, against water vapour and decreases Young’s modulus compared to samples containing the $\alpha^{\prime }$ crystal structure [20]. Guinalt et al. also reported that PLA containing α crystal structure was less permeable to oxygen than PLA containing only $\alpha^{\prime }$ crystal structure [21]. Direskens et al. have shown a correlation between the inner crystal structures and the morphological parameters of compression-molded Poly-d-l-Lactic Acid, PDLLA, (4:96%) and variation in oxygen transport characteristics (permeability, diffusivity, and solubility) [21]. They found that the α crystal structure has a strong impact on gas permeability and gas solubility. Schmid et al. [22] have reported a comparison between the water vapour transmission rate (WVTR) and oxygen transmission rate (OTR) of most popular plastics used for food packaging (Figure 1). The position of PLA on the graph shows that PLA has a better oxygen barrier property as compared to most synthetic polymers such as polystyrene (PS), low density polyethylene (LDPE), and high-density polyethylene (HDPE) as well as polymer materials that are used for packaging applications such as PP and PET. In their study, Toro et al. reported that the values of WVTR decrease linearly with increasing crystallinity of the PLA from 0% to 50% [23]. They prepared films from three different grades of commercial PLA with different ratios of l-lactide and d-lactide with crystallites ranging from 0% to 50%. They showed that Water Vapor Permeability (WVP) of PLA increased from 1 to 52 ${g\;\mathrm{mm}\;m}^{-2}d^{-1}$ (RH 90%) when increasing crystallinity from 0% to 50% [23].

Adding nucleating agents is an easy way of engineering the crystal structures within PLA. Pan et al. reported that the crystal structure of PLLA can be changed with the addition of poly (d- and l-lactide) (PDLLA). Furthermore, the incorporation of PDLLA (50 wt%) decreases the phase transition temperature from $\alpha^{\prime }$ to α structure and promotes the formation of α crystal structure in PLLA [14]. In another study, Lai et al. blended 3:2 4-dibenzylidene-d-sorbitol (DBS) with PLLA, which showed that the ordered and regular α crystal of PLLA was favored as the DBS was added. The formation of α crystals at lower temperatures is due to the π−π interaction of the DBS molecules that stack together to form a strand that was connected to PLA molecular chains via hydrogen bonding [24].

It was also shown by the same authors that PDLLA and DBS decrease the rate of crystallization of PLLA, and that DBS can act as a nucleating agent. Various types of nanoparticles have been used as nucleating agents for PLA-based materials. Specifically, the bio-resource nanoparticles have been shown to be an efficient way to overcome the low degree of crystallinity of neat PLA [25,26,27]. Nanoparticles such as cellulose nano-whiskers [25], chitosan nanoparticles [26], and starch nanocrystals [27] can improve the degree of crystallinity of PLA.

Among these mentioned nanoparticles, starch nanocrystals (SNCs) have several advantages such as renewability, high availability, biodegradability, and a reasonably large reactive surface [27,28,29,30]. The gas barrier properties of PLA/SNC nanocomposites strongly depend on dispersion of SNCs within the PLA matrix, their inherent properties (i.e., surface area), as well as the interfacial adhesion between SNCs and the PLA matrix. Furthermore, SNCs are known to modify crystalline properties and thus can alter the gas permeability of PLA [31]. Thus, with the addition of g-SNCs, the crystalline structure and the barrier properties of polymer films can be enhanced significantly.

For example, Yin et al. prepared a hydrophobic cross-linked starch nanocrystal (CSN)/PLA nanocomposite [32]. They showed that the SNC/PLA nanocomposite has better water vapour barrier properties due to the higher degree of crystallinity, which makes it difficult for the water molecule to penetrate. Muller et al., prepared a PLA/thermoplastic starch film by extrusion and thermo-pressing method [33], they showed that the water vapour barrier properties of the PLA were enhanced in the presence of thermoplastic starch. This is due to a decrease in the number of free hydroxyl groups present in the cross-linked starch chains, which hinders the diffusion of water molecules through the film matrix. The effect of the starch on the oxygen permeability of the PLA was further evaluated by Battegazzore et al., who prepared the maize starch/PLA blend with a co-rotating twin-screw micro extruder [34]. They showed that it was difficult for the oxygen molecule to penetrate the starch/PLA composites due to the formation of the clusters of starch, which can modify the intermolecular forces between the chains of the polymer and decrease the free volume [35].

However, to the best of our knowledge, no previous study has used grafted SNCs with lactic acid (g-SNCs) as a nucleating agent. The use of SNCs can improve α crystal structure content and thereby improve the gas barrier properties (i.e., oxygen and water vapour permeability) of PLA. Therefore, a comprehensive analysis of SNC content on the α crystal structure formation and their induced crystalline structure on the barrier performance of PLA is carried out in this work. In a previous study, we showed that the g-SNCs have a considerable effect on the kinetics of crystallization of PLA, in that it reduces the phase transition temperature of $\alpha^{\prime }$ to α crystal structure of PLA [36]. In this work, the effect of the different g-SNC nanoparticles loading on the crystalline formation of the α crystal structure and degree of crystallinity of PLA nanocomposites is quantified. Finally, the oxygen and water vapour permeability of PLA/g-SNC nanocomposites were evaluated. The results should unravel the role of g-SNCs and their induced crystalline structure on the barrier performance of polymer films.

## 2. Materials and Methods

### 2.1. Materials

The waxy corn starch used in this study mostly contains amylopectin, while only a trace of amylose was present. The waxy corn starch was purchased from Sigma-Aldrich Co., St. Louis, MO, USA. In addition, semi-crystalline commercial-grade PLA (4032 D) pellets with an l- to d-lactide ratio of 98:2 and a density of 1.24 g/cm^3^ were purchased from Nature Works LLC (Plymouth, MN, USA). The average molecular weight ($M_{w}$) and polydispersity index ($M_{w}/M_{n}$) of PLA pellets are about 109 kg/mol and 1.57 respectively.

### 2.2. Surface Modification of SNCs

Surface modification of SNCs was carried out as described in the previous work of the authors [34]. In summary, 30 g of starch nanoparticles were dispersed into 100 mL of tetrahydrofuran (THF) and the mixture underwent stirring to achieve homogeneous dispersion. Then, LA was added slowly to the stirring solution and the resulting suspension was heated up to 60 °C for 30 min under an argon atmosphere. The solution was left for THF evaporation overnight at room temperature. After THF evaporation, 150 mL of toluene was added to the reaction and the mixture was heated up to 85 °C for 24 h. After washing the resulting mixture with THF and ethyl acetate several times, the grafted starch nanoparticles were collected and vacuum dried at 45 °C.

### 2.3. Preparation of PLA/g-SNC Nanocomposite

PLA/g-SNC nanocomposites with different g-SNC content (i.e., 3, 5, and 7 wt%) were prepared by the casting process. Initially, PLA was dissolved in dichloromethane for about 10 to 15 min at 21 °C and after dissolving PLA, different concentrations of g-SNCs were added to the PLA solution. The solution was then stirred at room temperature for 24 h. Finally, the dissolved solution was poured into a petri dish, spread evenly, and then allowed to dry in a vacuum oven for about 24 h at room temperature in order to remove the traces of the dichloromethane solvent.

### 2.4. Methods

#### 2.4.1. Differential Scanning Calorimetry (DSC)

Differential scanning calorimetry (DSC) tests were performed using a TA instrument calorimetry (Q 2000) equipped with a liquid nitrogen cooling system. The appropriate amount of sample (~5–10 mg) was sealed in an aluminum pan and was heated from 25 to 200 °C at a heating rate of 5 °C/min and held for 5 min to eliminate any possible crystallinity or residual stress in the samples. Then, the samples were quenched to $T_{c}=130$ °C crystallization temperature and were allowed to crystallize for different crystallization times ranging from 10 to 60 min. Subsequently, the temperature was ramped back up to 200 °C with a heating rate of 10 °C/min in order to probe the melting behavior after recrystallization. In order to investigate the effect of g-SNCs on the degree of crystallinity ($X_{c}$), cold crystallization enthalpy ($\Delta H_{c}$) and, melting enthalpy ($\Delta H_{m}$) were determined from the second heating scan.

The $X_{c}$ was determined from a thermal effect accompanying melting or crystallization via the following formula:$$X_{c}=\frac{\Delta H_{m}}{\left(1-w\right)\Delta H_{0}}\times 100\%$$
where w is the weight fraction of sample and $\Delta H_{0}$ denotes the enthalpy of fusion of a 100% crystalline PLA (93.64 J g^−1^).

#### 2.4.2. Wide-Angle X-ray Diffraction (WAXD)

Wide-angle X-ray diffraction (WAXD) patterns were obtained using an X-ray diffractometer (D-8, Bruker, Billerica, MA, USA to detect the crystal structures of neat PLA and PLA nanocomposites after recrystallization at 130 °C for different crystallization times (10–60 min). The sample was exposed to an X-ray beam with the X-ray generators running at 40 kV and 40 mA. The copper *Kα* radiation (*k* = 1.542 Å) was selected, and the scanning was carried out at 0.03°/s in the angular region (2*θ*) of 5–40°. The thickness of the samples was around 0.3–0.5 mm. 

#### 2.4.3. Polarized Optical Microscopy (POM)

The effect of g-SNC nanoparticles on the morphology of spherulites of PLA was observed with polarized optical microscopy (Nikon 249171) in conjunction with the FP82HT hot stage (Mettler Toledo, Columbus, OH, USA) instrument. First, a small piece of the sample was cut and placed between two microscopy slides and pressed gently to form a thin film (thickness: ~20−50 µm). The specimen was then heated up to 200 °C on the hot stage and held at this temperature for 3 min to erase the thermal history. Then, it was cooled to a desired isothermal crystallization temperature (T_c_ = 130 °C) at 100 °C·min^−1^ and then held at the T_c_ to observe the spherulite growth rate. The growing spherulite crystals were monitored during solidification by taking photomicrographs at appropriate intervals. The spherulite diameter was measured with Image-Pro Plus 3.0 software. The hot stage is calibrated with a melting point standard with an accuracy of ±0.2 °C and dry nitrogen gas was purged throughout the hot stage during all the measurements and thermal treatments.

#### 2.4.4. Small Angle X-ray Scattering (SAXS)

Small angle X-ray scattering (SAXS) patterns of neat PLA and PLA nanocomposites were collected with a Bruker SAXS Nanostar system, equipped with a microfocus copper anode at 45 kV/0.65 mA, MONTEL OPTICS Geesthacht, Germany and a VANTEC 2000 2D detector, Billerica, Massachusetts, USA. The distance from the detector to the sample were calibrated with a silver behenate standard prior to the measurements. The scattering intensities were integrated from 0.10° to 2.80°. Collection exposure times were 500 s. The diffracted intensities were treated with Primus GNOM 3.0 programs from ATSAS 2.3 software EMBL, Hamburg, Germany, from which the blank was mathematically subtracted to determine the mean particle size and shape of the sample by pair distance distribution [28].

#### 2.4.5. Scanning Electron Microscopy (SEM)

The morphological changes of the samples were observed with scanning electron microscopy (S-7500, Hitachi, Tokyo, Japan). Before SEM characterization, samples were cut in liquid nitrogen, and they were coated with a thin layer of Pd-Au. The coating was performed using a vapour deposition process in order to avoid surface charging under an electron beam and to minimize sample damage by the electron beam.

#### 2.4.6. Oxygen Permeability (OP)

Oxygen permeability (OP) of neat PLA and PLA nanocomposites was measured according to the ASTM standard D3985-024 by Mocon Oxtran 2/21 (Brooklyn Park, MN, USA) oxygen permeability tester [12]. OTR measurements were carried out at 23 °C, at 0% relative humidity using high purity oxygen gas (>99.99%). The samples were then placed in a diffusion cell by purging nitrogen gas (>99.99% purity) for at least 24 h under an equilibrium humidity. During the test, pure oxygen at a pressure of 0.5 bar and at a rate of 20 mL·min^−1^ was introduced into the upper half of the sample chamber, while nitrogen gas was injected into the lower half of the chamber. The test was performed for at least 4 h, in order to attain a steady state. Oxygen permeability (m^3^·mm^−2^·s^−1^·Pa^−1^) was calculated by multiplying the OTR with the film thickness. All of the tests were repeated thrice and only the mean values are reported in the paper.

#### 2.4.7. Water Vapour Permeability (WVP)

Water vapour permeability (WVP) of PLA and PLA nanocomposite films was measured by a water vapour permeability test machine, PERMATRAN-W model 1 (Mocon, Brooklyn Park, MN, USA) following ASTM standard E398-03 [13]. The relative humidity (RH) was set to 100% in the wet chamber and 5% in the dry chamber, yielding a driving force of 95% RH. The circular sample having an area of 50 cm^2^ was tested in the chamber at atmospheric pressure and at a temperature of 25 °C. Tests were carried out until ten successive readings deviated less than 5% from their average value per sample. The resulting value was then reported as the WVTR value.

## 3. Results and Discussion

### 3.1. SEM Analysis

The SEM images from the cross-sectional structure of PLA/g-SNC nanocomposites with different g-SNC additions (3, 5, and 7 wt%) are shown in Figure 2. The surface of pure PLA was smooth and flat. When a small amount of g-SNC nanoparticles was added (3 wt%), the composite displayed a good dense cross-sectional structure, and the g-SNCs were found to be uniformly distributed within the PLA. When 5 wt% g-SNC was added, the g-SNCs were encapsulated well by the PLA with a uniform dispersion inside. When the g-SNC addition reached 7 wt%, some agglomerates of g-SNCs were present inside the PLA, revealing the rough cross-section of the nanocomposite. Agglomeration of g-SNCs at higher concentrations (7 wt%) can decrease active surface area or surface energy for nucleation and consequently reduce the rate of crystallinity (Section 3.2 and Section 3.3).

### 3.2. Melting Behavior of PLA/g-SNC Nanocomposites

It has been shown that the melting behavior of semicrystalline polymers such as PLA has a close relation with crystalline structures [21,22,23,24,25,26,27,28,29,30,31]. Thus, the effect of g-SNC content (ranging from 3 to 7 wt%) on the melting behavior of PLA nanocomposite was studied with the help of DSC measurements. In addition, the values of crystallinity ($X_{c}$) was measured and are listed in Table 1. Figure 3 shows the DSC thermograms of neat PLA and PLA/g-SNC nanocomposites, which are crystallized at 130 °C at different isothermal crystallization time (10–60 min). As can be shown in Figure 3a, no visible melting peak is observed for neat PLA that was crystallized for less than 10 min. However, as crystallization time increases to 30 min, two endothermic melting peaks are observed and the intensity of these two melting peaks is enhanced by increasing crystallization time. The formation of the dual melting peak is caused by the melting of poorly ordered PLA crystal structure ($\alpha^{\prime }$) followed by a simultaneous recrystallization and melting of more ordered PLA crystal structure (α) at higher temperature [21,22,23,24,25,26,27,28,29,30,31]. All PLA nanocomposites showed a dual melting peak except PLA/g-SNC (5 wt%) nanocomposite. This means that the α crystal structure is promoted in the presence of the 5 wt% g-SNC nanoparticles. In addition, the melting temperature of the PLA nanocomposite sample containing 5 wt% g-SNC was found to be considerably enhanced after 30 min. This implies that a 30 min period is sufficient for PLA to arrange its molecular chains in an orderly manner. In addition, the $X_{c}$ of the PLA/g-SNC nanocomposites increases when increasing the content of g-SNC nanoparticles up to 5 wt%, and then decreases with further increasing g-SNC nanoparticles content (Table 1). This could be due to the higher content of g-SNC nanoparticles (i.e., 7 wt%), which can suppress the molecular chain movement of PLA and therefore decrease the $X_{c}$. The maximum $X_{c}$ was obtained when adding 5 wt% g-SNC nanoparticles, which is 17.4% higher than that of a neat PLA sample. These results indicate that the crystallization capacity of neat PLA is relatively poor, while the addition of g-SNC nanoparticles enhances the crystallization of PLA to varying degrees. The lower degree of crystallinity can be attributed to some partial agglomeration of the g-SNCs at 7 wt% which was observed by SEM as well (Figure 2). The agglomeration reduces the surface energy for nucleation. Another reason for the lower crystallinity of PLA/g-SNC (7 wt%) nanocomposites could be that the lower chain motion of PLA in the presence of 7 wt% g-SNC is responsible for limited PLA short-chain arrangement into the α-PLA crystal structure.

### 3.3. Crystalline Structure of PLA/g-SNC Nanocomposites

In order to determine the effect of g-SNC content (varying from 3 to 7 wt%) on the crystalline structures of PLA nanocomposite, the WAXS measurements as a function of the crystallization time were performed (Figure 4). As can be seen from Figure 4a, a wide diffraction peak is observed in the case of the neat PLA sample, which was crystallized for less than 40 min. This is probably because 40 min is insufficient for the movement of the PLA molecular chains; therefore, it gives rise to a handful of short-range ordered lamellae. When isothermal crystallization time increases beyond 40 min, two strong diffraction peaks appear at 2θ=16.5° and 19.5°, which are attributed to the (110)/(200) and (203)/(113) planes respectively. In addition, three small diffraction peaks at 2θ=11.1°, 14.5°, and 22.5° were observed, which were assigned to the (010), (110)/(200) and (015) planes respectively.

These five diffraction peaks are characteristics of α crystal structure within the neat PLA [26]. Although all of these diffraction peaks appear in PLA/g-SNC nanocomposites, the crystallization time decreases considerably as g-SNC levels increase up to 5 wt%. The reduction in the crystallization time can be related to the nucleation ability of g-SNCs, which can promote the formation of α crystalline structure at lower crystallization time (20 min), when compared to neat PLA. However, a higher load of g-SNCs (7 wt%) in PLA restricts the formation of the PLA α crystals instead of promoting it, which results in an increase in the crystallization time to 30 min.

Moreover, WAXS measurement is also an effective way to determine the $X_{c}$ (Table 1). It is worth mentioning that the value of crystallinity obtained with WAXS analysis is slightly higher than the value of crystallinity obtained via DSC analysis. These differences can be attributed to the differences in the measurement principle [12]. Across the different PLA/g-SNC nanocomposites, the one with 5 wt% g-SNC showed the highest $X_{c}$. At a crystallization time of 0 min, PLA/g-SNC (5 wt%) nanocomposites were amorphous; however, after keeping the samples for 10 min at a crystallization temperature of 130 °C, a small number of crystals (with $X_{c}$ equal to 1%) were obtained. By enhancing the isothermal crystallization time to 20 min, PLA/g-SNC (5 wt%) exhibits a significant increase in $X_{c}$. In fact, the resulting value of $X_{c}$ was found to be 23.3%, which is a substantially higher $X_{c}$ than the neat PLA. This could be due to the presence of the g-SNC (5 wt%) nanoparticles that act as an efficient nucleating agent at a relatively lower isothermal crystallization time. Therefore, the crystallization rate of PLA is radically enhanced and more crystals are formed when g-SNCs were loaded at 5 wt%. In contrast, a relatively high load of g-SNCs (7 wt%) in PLA restricts the formation of the PLA α crystals, which decreases the value of $X_{c}$ to 13.4% [30].

### 3.4. Long Period (L_ac_) of PLA/g-SNC Nanocomposites

The SAXS characteristic provides valuable information on the average crystalline lamellar thickness of semicrystalline polymers such as PLA. Therefore, it can provide additional information about the contribution of the crystalline structures on the permeability of PLA/g-SNC nanocomposites [35,36]. The Lorentz-corrected SAXS profiles of neat PLA and PLA nanocomposite at $T_{c}$ = 130 °C for different crystallization times (10–60 min) were obtained and are shown in Figure 5. In addition, the morphological information, such as the long period ($L_{ac}$), the crystal layer thickness ($L_{c}$), and the amorphous layer thickness ($L_{a}$ = $L_{ac}-L_{c}$), are obtained from a one-dimensional correlation function [37,38]. The data hence obtained are summarized in Table 2 [39,40]. As can be shown in Figure 5a, even after 40 min of crystallization time, no scattering peak was observed in the SAXS scattering pattern for neat PLA. However, when crystallization time is equal to 50 min, a scattering peak appears at the scattering vector (q) of 0.30 nm, which corresponds to the $L_{ac}:$ ~20.9 nm. At a higher crystallization time (60 min), q shifts to a lower value (0.28 nm). This indicates that increasing crystallization time decreases the q value. Although a similar SAXS profile is observed in the case of PLA/g-SNC nanocomposites, the scattering peaks become visible at a lower crystallization time. For instance, a clear scattering peak is observed in the case of PLA/g-SNC (5 wt%) nanocomposite which is crystallized for 20 min and the position of q is shifted to a lower value when increasing the crystallization time (Table 2). This decrease in crystallization time is clearly affected by the morphological parameters of PLA nanocomposites. As can be seen in Table 2, PLA/g-SNC (5 wt%) which were crystallized for 20 min show a very weak scattering peak with q equal to 0.33 nm, corresponding to an $L_{ac}$ of ~19.9 nm. However, this scatting peak is absent in the both PLA/g-SNC (3 wt%) and PLA/g-SNC (7 wt%) nanocomposite samples which were crystallized for the same crystallization time. The absence of the SAXS signals in the PLA/g-SNC nanocomposite at low concentration of g-SNCs (3 wt%) can be due to the less active surface energy for crystallization.

This can lead to a longer induction time for crystallinity of PLA. In contrast, higher concentrations of g-SNCs (7 wt%) notably impede the motion of the PLA chains and limits their ordered arrangement into a short- and long-range ordered structure.

In addition, the inclusion of 5 wt% of g-SNCs into PLA results in a lower $L_{a}$ and a higher $L_{ac}$, resulting from an increase in $L_{c}$. However, in PLA/g-SNCs (7 wt%), despite the same $L_{ac}$ as that of a smaller$\;L_{c}$, higher $L_{a}$ values are achieved. This further confirms that a high content of g-SNCs greatly suppress the ordered arrangement of PLA chains, giving rise to the loose and less perfect PLA spherulite, as can be seen in Figure 6. The smaller$\;L_{c}$ and higher $L_{a}$ can be attributed to lower short- and long-chain motion of PLA molecular chains due to higher agglomeration of g-SNCs (7 wt%), as is observed by SEM (Figure 2).

### 3.5. Spherulite Morphology of PLA/g-SNC Nanocomposites

Apart from the crystalline structures, the spherulite morphology is another important factor that can help to better understand the gas permeability of PLA. Figure 6 shows the evaluation of the spherulite morphology of neat PLA and PLA/g-SNC nanocomposites during isothermal crystallization at 130 °C. It can be seen that more PLA spherulites are formed in all PLA nanocomposites than in neat PLA at the same isothermal crystallization time. This indicates that the evaluation of α crystal density in PLA nanocomposites is higher than that of neat PLA. However, the g-SNCs have a different effect on the spherulite morphology based on the concentrations. After crystallization at 130 °C for 10 min in the case of the PLA/g-SNC (3 wt%) nanocomposite, only a small number of spherulites were observed because only a small number of g-SNC nanoparticles are present to act as a heterogeneous surface area for the growth of the PLA molecular chain. However, for PLA with 5 wt% g-SNC that crystallized in the same crystallization time (10 min), a higher number of small, and denser spherulites was observed. This is due to the excellent heterogeneous nucleating ability of g-SNC nanoparticles, which effectively increases the α nucleating density and spherulite growth of PLA. In contrast, the addition of 7 wt% g-SNC slows the motion of PLA chains substantially and restricts their relaxation. The critical concentration of g-SNC nanoparticles to form a network structure in the PLA matrix was therefore estimated to be about 5 wt%. It is well documented that the overall crystallization performance depends on interactions of thermodynamic, kinetics, and transport processes. Therefore, it is logical to conject that the addition of 5 wt% g-SNC acts as an efficient α nucleating agent, exhibiting a better dispersion during isothermal crystallization. In addition, the thickness of the interlamellar amorphous layer calculated by the one-dimensional electron density correlation function is less than 20 nm, much smaller than the size of g-SNCs (ranging from 80 to 120 nm). Thus, most of the g-SNCs are preferentially dispersed between PLA spherulites.

### 3.6. Gas Transport Properties of PLA/g-SNC Nanocomposites

The effect of g-SNC content on the OP (23 °C and 0% RH) and WVP (23 °C and 100% RH) permeability of PLA and PLA/g-SNC nanocomposites was determined using an oxygen and water permeability machine and the results are shown in Figure 7a,b respectively. As can be seen, the OP and WVP of all PLA nanocomposites is lower than neat PLA. Specifically, a considerable reduction in the gas permeability of PLA/g-SNC (5 wt%) was observed. In fact, the OP was reduced by a factor of two, whereas the WVP was reduced by 70% compared to that of neat PLA. This improvement in the gas barrier performance of the PLA/g-SNC nanocomposites can be attributed to the presence of highly crystalline g-SNCs, which act as efficient nucleating agents and result in a higher degree of crystallinity. It is believed that the permeate molecule does not readily diffuse through crystallites [19]. Another plausible explanation for the enhancement of the oxygen and water vapour barrier properties of PLA/g-SNC nanocomposites can be attributed to an increase in the effective travel path of oxygen and water vapour that makes the path of water vapour and oxygen more tortuous, resulting in a reduction in gas diffusion and hence lower gas permeability. Among the PLA/g-SNC nanocomposites tested, the OP and WVP of PLA/g-SNC (5 wt%) were minimum. As discussed in the WAXS and DSC results, the addition of 5 wt% g-SNC leads to more formation of impermeable α crystalline structure as well as a higher degree of crystallinity [6,7,8,9,10,11,12,13,14,15,16,17,18,19]. Although the addition of 3 and 7 wt% g-SNC did improve the gas barrier properties of PLA, the results showed lower gas barrier properties compared to the PLA/g-SNC (5 wt%) nanocomposite. The reduction in gas barrier properties could be due to the presence of more of the amorphous phase rather than the crystalline phase and/or formation of more $\alpha^{\prime }$ crystalline that is more permeable than α crystalline structure in these PLA nanocomposites. In addition, the L_c_ and L_a_ of PLA nanocomposites with 7 wt% of g-SNC nanoparticles had lower and higher values, respectively, compared to PLA/g-SNC (5 wt%) despite having the same L_ac_ (Table 2).

In conclusion, the maximum increase in crystallinity occurred at 5 wt% g-SNC content. This correlates well with the optimum improvement in the gas barrier properties of the films. It should be mentioned that the addition of more than 7 wt% g-SNC did not improve the crystallinity of PLA further, probably due to retarded crystal growth. This was caused by g-SNC agglomerations at high loading levels (kinetic restriction), which resulted in lower OP and WVP. Similarly, the addition of 3 wt% g-SNC did not improve the crystallinity of PLA because it has lower surface free energy and higher thermodynamic barriers [41,42].

## 4. Conclusions

In this work, the effects of g-SNC content (3–7 wt%) on the α crystal structure and WVP and OP of PLA were investigated. All samples were isothermally crystallized at 130 °C for different crystallization times and α crystal structure was observed. It was found that the g-SNC nanoparticles decreased the time of evaluation of α crystal structure and increased the content of α crystal structure. Therefore, it seems that g-SNC nanoparticles act as a α nucleating agent for PLA. In addition, the PLA nanocomposite film had a better barrier to both water and oxygen molecules, irrespective of g-SNC content. This is attributed to the presence of highly crystalline g-SNCs, which increased the degree of crystallinity of the PLA nanocomposites and act as impermeable regions in the PLA matrix. Finally, the maximum improvements in water vapour permeability (50%) and oxygen permeability (70%) of PLA nanocomposites was achieved when 5 wt% g-SNC nanoparticles were added into PLA matrix. However, higher content of the g-SNC nanoparticles (7 wt%) had a negative effect on the crystallinity and, consequently, on the barrier properties. To conclude, PLA/g-SNC nanocomposite films with better barrier performance were developed in this study, and they have tremendous potential in food packaging applications that are looking for biodegradable and renewable alternatives.

## Figures and Tables

**Figure 1 polymers-14-02802-f001:**
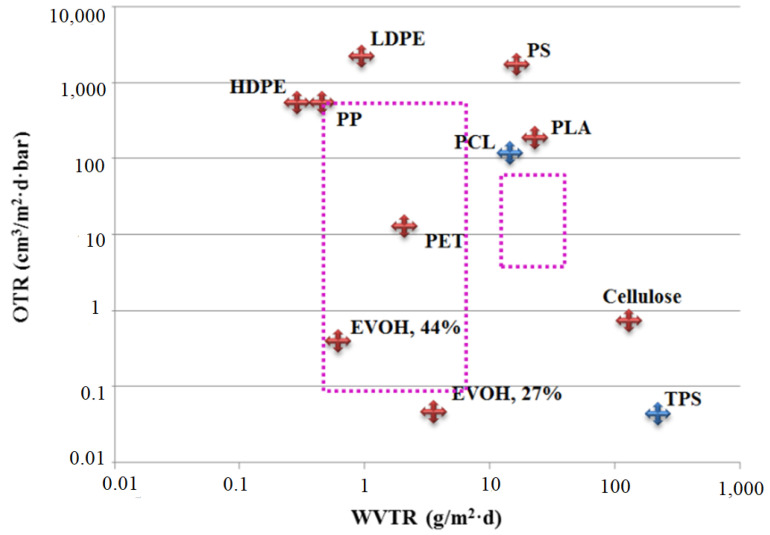
Comparison between barrier requirements of some food (dotted lines) and barrier properties of some plastics used in food packaging. WVTR and OTR values of materials are normalized at 100 µm of thickness. Reprinted with permission from Ref. [23]. Copyright 2017 Copyright Elsevier.

**Figure 2 polymers-14-02802-f002:**
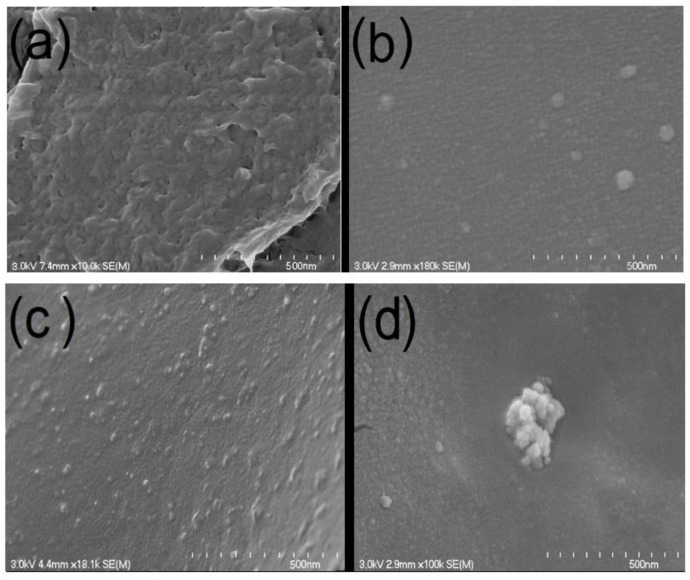
SEM images of: (**a**) neat PLA, (**b**) PLA/g-SNC (3 wt%), (**c**) PLA/g-SNC (5 wt%), and (**d**) PLA/g-SNC (7 wt%) nanocomposites.

**Figure 3 polymers-14-02802-f003:**
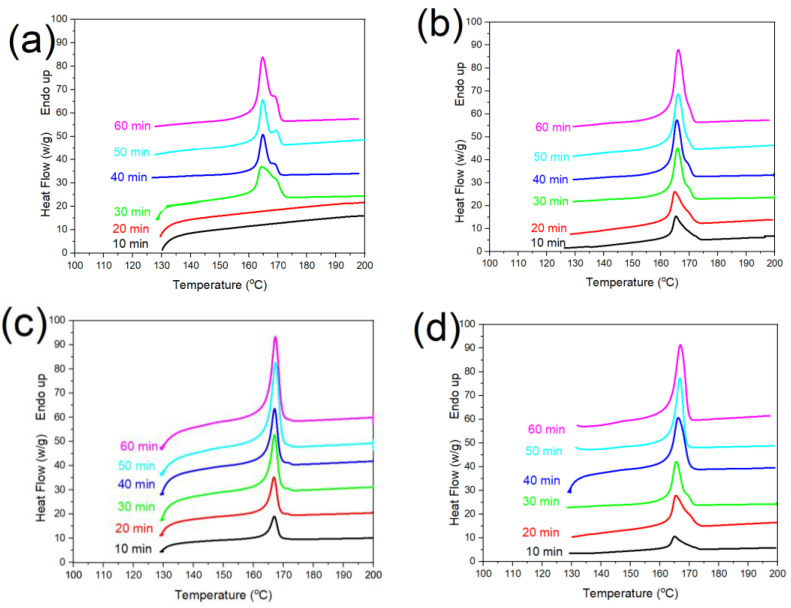
The melting behavior of (**a**) neat PLA, (**b**) PLA/g-SNC (3 wt%), (**c**) PLA/g-SNC (5 wt%), and (**d**) PLA/g-SNC (7 wt%), nanocomposites, which are crystallized for different crystallization times (10–60 min) at $T_{c\;}$ = 130 °C.

**Figure 4 polymers-14-02802-f004:**
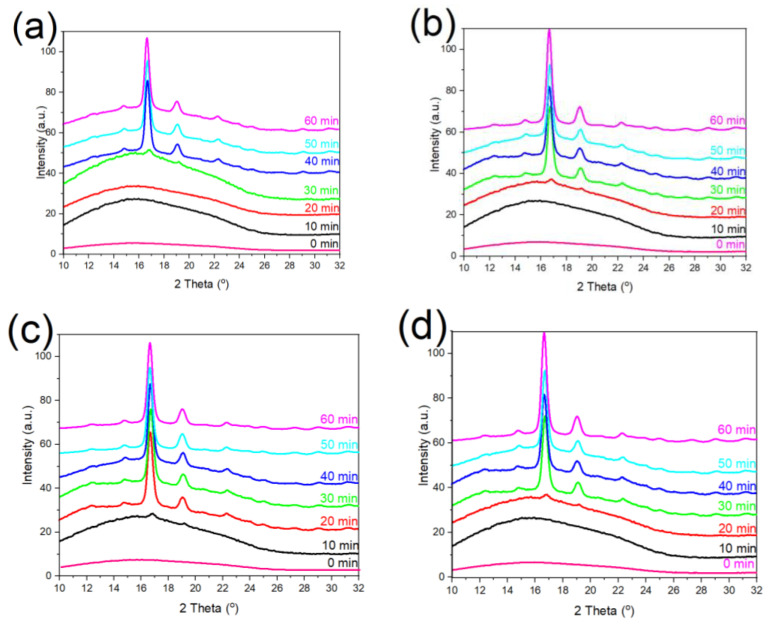
The WAXS analysis of (**a**) neat PLA, (**b**) PLA/g-SNC (3 wt%), (**c**) PLA/g-SNC (5 wt%), and (**d**) PLA/g-SNC (7 wt%), nanocomposites, which are crystallized for different crystallization times (10–60 min) at $T_{c\;}$ = 130 °C.

**Figure 5 polymers-14-02802-f005:**
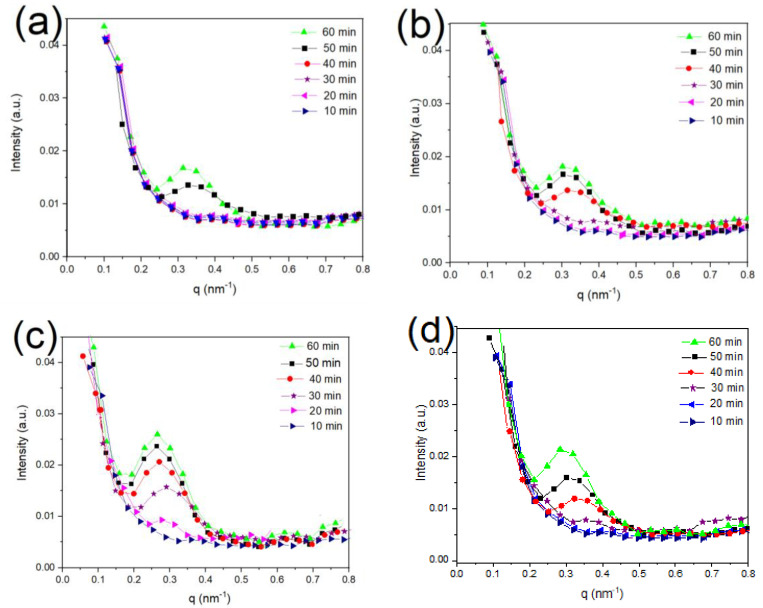
The SAXS analysis of (**a**) neat PLA, (**b**) PLA/g-SNC (3 wt%), (**c**) PLA/g-SNC (5 wt%), and (**d**) PLA/g-SNC (7 wt%) nanocomposites which are crystallized for different crystallization times at $T_{c\;}$ = 130 °C.

**Figure 6 polymers-14-02802-f006:**
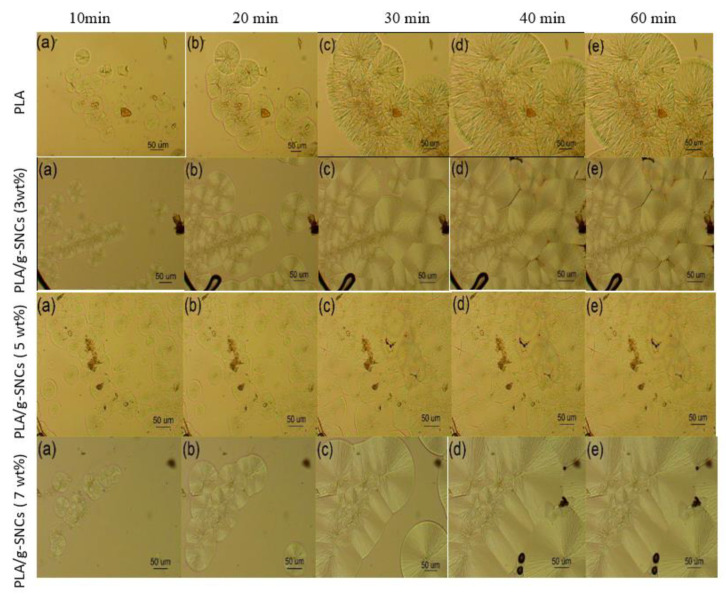
Optical microscopic images of neat PLA, PLA/SNC-g-LA (3 wt%), PLA/SNC-g-LA (5 wt%) and PLA/SNC-g-LA (7 wt%) nanocomposites at (**a**) 10 min, (**b**) 20 min, (**c**) 30 min, (**d**) 40 min, and (**e**) 60 min.

**Figure 7 polymers-14-02802-f007:**
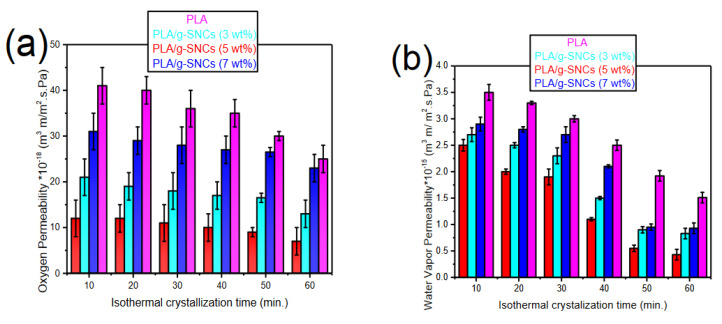
(**a**) Oxygen and (**b**) water vapour permeability of neat PLA and PLA/SNC-g-LA nanocomposites crystallized at 130 °C at various crystallization time.

**Table 1 polymers-14-02802-t001:** DSC and WAXS data of PLA and PLA/g-SNC nanocomposite, which are crystallized for different crystallization times (10–60 min) at $T_{c}\;$ = 130 °C.

Sample	PLA				PLA/g-SNC (3 wt%)			PLA/g-SNC (5 wt%)			PLA/g-SNC (7 wt%)		
Time (min)	$T_{m1}$ (°C)	$T_{m2}$ (°C)	$\chi_{WAXS}$ (%)	$\chi_{DSC}$ (%)	$T_{m2}$ (°C)	$\chi_{WAXS}$ (%)	$\chi_{DSC}$ (%)	$T_{m1}$ (°C)	$\chi_{WAXS}$ (%)	$\chi_{DSC}$ (%)	$T_{m2}$ (°C)	$\chi_{WAXS}$ (%)	$\chi_{DSC}$ (%)
10	-	-	-	-	165.6	-	1.00	166.1	-	4.20	166.5	-	1.20
20	-	-	-	-	165.8	-	10.3	167.4	24.9	23.3	167.6	-	13.4
30	169.2	167.3	-	10.7	168.7	21.9	15.0	168.9	37.5	37.0	167.7	33.9	17.0
40	165.5	168.3	-	12.5	168.9	22.7	25.5	169.2	38.9	38.0	168.1	34.7	27.5
50	164.4	167.4	-	15.7	168.3	35.5	32.1	169.0	39.0	38.9	168.5	36.5	31.1
60	164.1	167.7	35.2	17.1	169.5	36.3	33.4	169.1	41.8	41.1	168.6	37.8	32.4

**Table 2 polymers-14-02802-t002:** SAXS data of PLA and PLA/g-SNC nanocomposite which are crystallized at 130 °C and at various crystallization times.

	PLA			PLA/g-SNC (3 wt%)			PLA/g-SNC (5 wt%)			PLA/g-SNC (7 wt%)		
Time	$L_{ac}$	$L_{c}$	$L_{a}$	$L_{ac}$	$L_{c}$	$L_{a}$	$L_{ac}$	$L_{c}$	$L_{a}$	$L_{ac}$	$L_{c}$	$L_{a}$
20	-	-	-	-	-	-	19.0	14.0	5.0	-	-	-
30	-	-	-	-	-	-	19.9	15.1	4.8	-	-	-
40	-	-	-	20.9	14.9	6.0	20.9	15.4	5.5	20.9	15.0	5.9
50	20.9	14.8	6.1	21.7	15.5	6.2	22.4	16.4	6.0	21.7	15.4	6.3
60	21.7	15.6	6.1	21.9	15.9	6.0	22.4	16.5	5.9	22.1	16.1	6.0

Scale of ${\;L}_{ac}$,${\;L}_{c}{\;\mathrm{and}\;L}_{a}$ are nm.

## Data Availability

Not applicable.

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
