# Peer review of "Crystallinity and Gas Permeability of Poly (Lactic Acid)/Starch Nanocrystal Nanocomposite"

_polymers, 2022, doi:10.3390/polym14142802_

Round 1

Reviewer 1 Report

Manuscript ID: polymers-1776524 – the revion of 1728701 submission
Title: Crystallinity and Gas permeability of Poly (lactic acid)/ Starch nanocrystals Nanocomposite

Authors: Somayeh Sharafi Zamir1,2*, Babak Fathi1, Abdellah Ajji3, Mathieu Robert1 and Said Elkoun

Authors described blending of commercial PLA with starch nanocrystals in order to increase its crystallinity degree for improvement of gas barrier properties. Crystallinity degree and quality of crystals were examined by DSC, SAXS and WAXS and optical microscopy.

First of all, no detailed answers to reviewers comments were added to the submission. It is then harder to find places, where authors improved current submission/revision. This should editor control.

Here, rather shortening of the related paragraphs can be considered. There is a lot of similar or the same explanations and used words in results explanation obtained by different techniques. The text is thus longer as is necessary.

Some new paragraphs appeared in the text, however, no consideration of my previous comments related to shortening of simple discussed results was done and answered. Here, editor should decide, if the article is worth to publish in the current rather long version.

New paragraph related to literature studies defines WTR and OTR. These abbreviations must be unified in the whole text. Related Figure 1 has to be checked if this figure is published with the permission of reference or was modified.

 Comments:

·       new sentences related to pi-pi stacking of sorbitol molecules should be better English edited.

·       SEM analysis was added prior to other analyses and possible agglomeration of SNC at higher 7 wt% concentration was discussed. (on the wish of the reviewer) However, this result stand as alone without trying to incorporate it into the text and discuss with the other results. For example, into the text related to WAXS analysis, some interconnection with SEM conclusions why 7 wt% SNC are less crystalline could be placed. Similarly, in paragraph 3.4, where lower chain motion is responsible for limited ordering arrangement to short and long range due to interaction with the higher conc of SNC. Also here, lower surface area of aggregates can be considered as another effect of this behavior.

In conclusion, revision was done in order to improve quality of the work. However, there is a space to do it better before the publication.

Reviewer 2 Report

Dear,

The authors developed PLA/g-SNCs nanocomposites, aiming to study crystallinity and barrier. t is a good manuscript to expand the literature database. The weak point of the manuscript was the process using solvent, which limits its application to food packaging. The production of nanocomposites in a film extruder would be closer to reality. In addition, the improvements below must be accommodated in the manuscript before publication:

1°) At the end of the abstract, suggest an application of the developed materials;

2º) Page 2. “It has been shown that the gas barrier properties of semicrystalline polymers such as PLA are substantially modified by crystalline polymorphisms”. Which studies? Please add references.

3º) The introduction needs to be updated with new references. I recommend updating with references from 2015 to 2022.

4º) Is the density unit (1.24 g/cc) correct? Wouldn't it be g/cm3?

5º) DSC. “The appropriate amount of sample (~ 5-100 mg)....” Is this really correct? Wouldn't it be 10 mg? Did the authors not use a precision balance to fix the exact value?

6º) Page 4. “In order to investigate the effect of g-SNCs on the degree of crystallinity (??), cold crystallization enthalpy (∆??) and, melting enthalpy (∆??) were determined from the first heating scan.” In general, the data from the second heating run is used, since it eliminates the thermal history of the sample. Why did the authors adopt the first cycle? It wasn't clear.

7º) WAXD. How thick were the samples? Please add to manuscript.

8º) Page 6. Please standardize terms in the manuscript. Use nanocomposites or composites.

9°) Table 1. How was the degree of crystallinity determined? Please add the equation to the methodology.

10°) The development of PLA films using toxic solvent is not suitable for packaging, due to migration and contamination. The authors should try to produce the same nanocomposites in film extruders.

Reviewer 3 Report

why is some text highlighted in red? should we read this text in some special way?

Lack of numbering of ropes - this significantly hinders reviewing

lines 3-4- (2.2. Surface modification of SNCs suggests repeating a sentence of two same words "dispersion"

graphs from 3-5 should have numerical values on the oY axes

Round 2

Reviewer 2 Report

Dear,

The authors satisfactorily answered the questions. The manuscript has been improved with the addition of revisions and suggestions.

Yours sincerely,